# Delayed Comprehensive Stroke Unit Care Attributable to the Evolution of Infection Protection Measures across Two Consecutive Waves of the COVID-19 Pandemic

**DOI:** 10.3390/life11070710

**Published:** 2021-07-19

**Authors:** Annahita Sedghi, Timo Siepmann, Lars-Peder Pallesen, Heinz Reichmann, Volker Puetz, Jessica Barlinn, Kristian Barlinn

**Affiliations:** Department of Neurology, University Hospital Carl Gustav Carus, 01307 Dresden, Germany; timo.siepmann@ukdd.de (T.S.); lars-peder.pallesen@ukdd.de (L.-P.P.); heinz.reichmann@ukdd.de (H.R.); volker.puetz@ukdd.de (V.P.); jessica.barlinn@ukdd.de (J.B.); kristian.barlinn@ukdd.de (K.B.)

**Keywords:** stroke care, stroke unit care, code stroke, protected code stroke, COVID-19, pandemic

## Abstract

We aimed to assess how evidence-based stroke care changed over the two waves of the COVID-19 pandemic. We analyzed acute stroke patients admitted to a tertiary care hospital in Germany during the first (2 March 2020–9 June 2020) and second (23 September 2020–31 December 2020, 100 days each) infection waves. Stroke care performance indicators were compared among waves. A 25.2% decline of acute stroke admissions was noted during the second (n = 249) compared with the first (n = 333) wave of the pandemic. Patients were more frequently tested SARS-CoV-2 positive during the second than the first wave (11 (4.4%) vs. 0; *p* < 0.001). There were no differences in rates of reperfusion therapies (37% vs. 36.5%; *p* = 1.0) or treatment process times (*p* > 0.05). However, stroke unit access was more frequently delayed (17 (6.8%) vs. 5 (1.5%); *p* = 0.001), and hospitalization until inpatient rehabilitation was longer (20 (1, 27) vs. 12 (8, 17) days; *p* < 0.0001) during the second compared with the first pandemic wave. Clinical severity, stroke etiology, appropriate secondary prevention medication, and discharge disposition were comparable among both waves. Infection control measures may adversely affect access to stroke unit care and extend hospitalization, while performance indicators of hyperacute stroke care seem to be untainted.

## 1. Introduction

While an exceeding number of patients infected with severe acute respiratory syndrome-Coronavirus-2 (SARS-CoV-2) led to relocation of in-hospital resources, time-sensitive treatment of acute stroke needs to be ensured [1]. In consequence, stroke care frequently demands overlapping resources in coronavirus disease 2019 (COVID-19), overwhelming hospitals. Protected code stroke protocols were established globally during the first wave of the pandemic to maintain timely access to hyperacute interventions, such as intravenous thrombolysis (IVT) and endovascular therapy (EVT), ensure safety of patients, as well as healthcare workers, and preserve best possible in-hospital acute care for patients infected with SARS-CoV-2 [2,3]. However, due to evolving in-hospital cohorting and isolation strategies, stroke patients infected with SARS-CoV-2 are largely treated in designated COVID-19 units not primarily specialized in evidence-based stroke care. While multiple studies suggest a global decline in stroke admission and acute intervention rates [4,5,6], there is sparse of data on how many patients were eventually deprived of in-hospital evidence-based stroke care due to the pandemic.

To reflect increasing infection control measures established in the first and intensified during the further course of the pandemic, we aimed to investigate their impact on in-hospital evidence-based stroke care by comparing stroke admissions during the two waves of the pandemic in a severely SARS-CoV-2 affected region in Germany.

## 2. Materials and Methods

### 2.1. Study Design and Population

We retrospectively analyzed consecutive code stroke patients who presented to the emergency department of a tertiary care hospital in Saxony, Germany, during the first and second waves of the pandemic. According to data provided by the national infection control center (i.e., Robert Koch Institute), the first wave approximately lasted from 2 March to 9 June 2020 [7]. The second wave was ongoing at the time of the analysis; however, for reasons of comparability of the two study cohorts, a respective 100-day observational period was chosen for final analysis. Consequently, the second wave was investigated from 23 September to 31 December 2020. The study was approved by the institutional review board (IRB) of Technische Universitaet Dresden (BO-EK-154042020). Due to the observational nature of the study design, informed consent was waived.

During the first wave of the pandemic, code stroke patients were regularly admitted to the hospital’s general emergency department and immediately seen by a stroke neurologist (Appendix A). Upon arrival, patients routinely underwent infection screen for respiratory or other symptoms suggestive of COVID-19 (Appendix B). If initial infection screen was positive, laboratory testing for SARS-CoV-2 using real-time reverse-transcription polymerase chain reaction (RT-PCR) from oropharyngeal swab was performed. From 2 April onwards, RT-PCR for SARS-CoV-2 was performed routinely in all patients. Protected hygienic measures, such as patient isolation and use of personal protective equipment, were maintained in positively screened patients until test results were available, up to 12 h following testing [8]. Meanwhile, patients were admitted to the comprehensive stroke unit or neurological intensive care unit (following standard-of-care EVT and IVT, if indicated). If test results were positive for SARS-CoV-2, patients were secondarily transferred to a designated COVID-19 isolation ward and treated by inpatient neurology consultation service. During the second wave of the pandemic, intensified infection control measures were implemented to contain the spread of COVID-19 more effectively. In detail, code stroke patients referred by emergency medical service or transferred from outside hospitals underwent pre-admission infection control screen. If the infection screen was indicative of possible COVID-19, patients were primarily admitted to a designated COVID-19 emergency ward run by internal medicine, where further infection control measures, such as rapid antigen and RT-PCR testing, for SARS-CoV-2, as well as protective isolation, were conducted. Stroke consultation service was provided to these patients throughout isolation. Secondary transfer to the respective neurological ward was initiated once COVID-19 status was judged non-infectious, as indicated by an RT-PCR cycle threshold >30. All other code stroke patients, including those potentially amenable to acute recanalization therapies, were primarily admitted to the general emergency ward without modifications to acute stroke treatment standards and secondarily transferred to either neurological or COVID-19 isolation ward, according to their infection screening and rapid antigen SARS-CoV-2 test results.

### 2.2. Data Acquisition

Patients were identified via a central data query applying respective ICD-10-GM-2021 codes (Appendix A). Patient data was extracted manually from all available sources, including the hospitals electronic patient database management and information system, admission, follow-up, and discharge summaries, as well as discharge summaries provided by rehabilitation centers and other hospitals. Data included demographical information, medical history, diagnoses, and treatment, as well as characterization of stroke etiology, phenotype, and severity, according to the National Institutes of Health Stroke Scale (NIHSS) and modified Rankin Scale (mRS) scores.

### 2.3. Performance Indicators of Evidence-Based Stroke Care

We assessed the proportion of patients who underwent acute reperfusion therapy consisting of IVT (administered either at the stroke center or at outside hospital prior to patient transfer) and EVT, as well as revascularization therapies, including acute carotid artery endarterectomy or carotid artery stenting. We also assessed frequency of patients undergoing immediate (i.e., at the day of admission) and delayed (i.e., >24 h following admission) stroke unit admission following emergency department presentation and acute recanalization therapies. We detailed pre- and in-hospital process times of hyperacute stroke care comprising onset-to-door, door-to-imaging, door-to-needle, door-to-groin, and onset-to-groin times. According to international stroke guidelines, secondary prevention medication, including antiplatelet and anticoagulant, use for ischemic stroke was evaluated for appropriateness [9]. We further analyzed length of hospitalization, and discharge disposition, including necessity of institutional care, as well as clinical and functional outcomes at discharge, using the NIHSS and mRS scores. Neurological deterioration was defined as any worsening of the NIHSS score during hospital stay.

### 2.4. Statistical Analysis

Descriptive analysis was used to describe categorical and continuous data. For continuous data, normality was checked descriptively (skewness, kurtosis), as well as analytically (Shapiro-Wilk test). Non-normally distributed continuous data was summarized using median and interquartile range (IQR). Normally distributed continuous data was summarized using mean ± standard deviation (SD). Categorical data was specified using frequency and percentages. Parametric analysis was applied for normal distributed continuous data, as well as non-normal distributed continuous data, in cases where the number of observations exceeded 50, accounting for the central limit theorem. Between-group comparisons between both waves were performed using Mann–Whitney U test in case of ordinal and non-normally distributed continuous data with an observation count less than 50. In case of dichotomous data, Fisher exact test was applied. Unpaired t-test was used for between-group comparisons of continuous data and non-normally distributed continuous data with an observation count greater than 50. A two-sided significance level alpha of 0.05 was corrected for multiple comparisons between performance indicators of evidence-based stroke care using conservative Bonferroni correction (factor 16), resulting in a *p*-value < 0.003 to be considered significant. Incidence-rate-ratio (IRR) was calculated by dividing the number of admissions during the second wave by the number of admissions during the first wave of the pandemic. Ninety-five percent confidence interval (95%CI) was computed by using the Wald method. Dataset was analyzed in an available case analysis approach, and missing values were not imputed. Amount of missing data was reported were applicable. Analysis was done using Stata^®^ Release 17 (StataCorp, 2021, College Station, TX, USA: StataCorp, LLC.).

## 3. Results

During the observational periods, a total of 878 code stroke patients were admitted to the emergency department. Of these, 582 (66.3%) were eventually diagnosed with transient ischemic attack (n = 72), acute ischemic stroke (n = 460), or intracerebral hemorrhage (n = 50). Overall, there was a 25.2% drop in total number of stroke patients presenting during the second wave (n = 249) as compared to the first (n = 333) wave of the pandemic, corresponding to an IRR of 0.75 (95%CI, 0.69–0.79) (Figure 1).

Median age (77 (65, 83) vs. 78 (65, 84)), as well as proportion of females (45.7% vs. 43.8%), did not differ between groups (*p* = 0.65 and *p* = 0.65). Patients admitted to hospital during the first wave were as severely affected of stroke as patients admitted to hospital during the second wave of the pandemic, reflected by similar baseline NIHSS scores (4 (2, 14) vs. 5 (2, 16), *p* = 0.36) and displayed a comparable patient history for all cardiovascular risk factors and comorbidities investigated (Table 1). Admission modalities encompassing admission via emergency medical service, intra- or inter-hospital transfer, following telestroke consultation or walk-in did not differ between groups (*p* > 0.05). While 11/249 (4.4%) patients were tested positive for SARS-CoV-2 in the second wave, none of the patients admitted during the first wave of the pandemic were found positive (*p* < 0.001). Table 1 details baseline characteristics of both study cohorts.

### Evidence-Based Stroke Care

The proportion of patients undergoing acute recanalization therapies, including IVT and EVT, did not differ between the two waves (122 (36.5%) vs. 92 (37.0%), *p* = 1.0). Moreover, process times of hyperacute stroke care, including onset-to-admission time in case of presentation via emergency medical service (90 vs. 132 min., *p* = 0.98), door-to-imaging time in case of acute therapy (9 vs. 12 min., *p* = 0.37), as well as door-to-needle time (38.4 vs. 37.8 min., *p* = 0.11), door-to-groin time (64.8 vs. 70.2 min., *p* = 0.08), and onset-to-groin time (234 vs. 238.8 min., *p* = 0.70), were similar in both waves. No differences with regard to acute revascularization therapies were evident among patients admitted during the first and the second waves of the pandemic (25 (7.5%) vs. 13 (5.2), *p* = 0.31). Table 2 shows clinical stroke management and outcome parameters. Further details on vascular phenotypes, including stroke etiology and severity, are shown in Table 3.

There was a larger proportion of acute stroke patients who did not receive stroke unit care within the same day of admission during the second compared with the first wave (17 (6.8%) vs. 5 (1.5%); *p* = 0.001). While none of the patients were tested positive for SARS-Cov-2 in the first wave of the pandemic, reasons for delayed stroke unit admission solely comprised limited bed capacities (n = 2) or medical emergencies (n = 3). Of 17 patients whose stroke unit admission was delayed during the second wave, ten (59%) were eventually tested SARS-CoV-2 positive accounting for 91% of all SARS-CoV-2 positive stroke patients in the second wave. One patient was tested SARS-CoV-2 positive after completion of stroke unit treatment. Reasons for delayed stroke unit admission in the remainder seven (41%) patients were suspicion of COVID-19 (n = 2), limited bed capacities (n = 4), and medical issues (n = 1).

Overall length of hospitalization was prolonged during the second wave of the pandemic when compared to the first wave (7 (4, 13) vs. 5 (5, 17) days, *p* < 0.001). When we analyzed length of hospitalization by discharge modality, results remained significant solely for patients transferred to an inpatient rehabilitation facility (20 (12, 27) vs. 12 (8, 18) days, *p* < 0.0001). Finally, analysis of medication at discharge revealed a comparable distribution of secondary preventive platelet inhibition and anticoagulant use for ischemic stroke among both infection waves (Table 2). On an individual patient basis, no deviations from national and institutional guidelines were registered (data not shown).

At discharge, patients displayed a tendency toward a higher degree of disability during the second wave, as reflected by NIHSS (2 (0, 7) vs. 3 (0, 11), *p* = 0.041) and mRS scores (2 (1, 4) vs. 3 (1, 4), *p* = 0.02), which remained non-significant after correcting for multiple comparisons. Discharge modalities encompassing discharge to home, a rehabilitation center or a care facility, or the transfer to a different department or hospital or death were equally distributed among patients admitted during the first and the second wave of the pandemic (Table 2). There was no difference regarding frequency of neurological deterioration in either wave of the pandemic (69 (20.9%) vs. 53 (21.4%), *p* = 0.18). COVID-19 patients more frequently experienced neurological deterioration during hospital stays as compared with non-COVID-19 patients (4 (36.4%) vs. 82 (18.7%)); however, this observation did not achieve statistical significance (*p* = 0.14). A detailed description of the amount of missing data is provided in Appendix C (Table A1).

## 4. Discussion

The major findings of our study are (1) a 25% decline in stroke admissions between the first and the second wave of the pandemic was paralleled by an increase of regional SARS-CoV-2 incidence in a COVID-19 epicenter region in Germany; (2) hyperacute stroke care did not appear to be affected by intensified infection control practices, yet isolation and cohorting measures had an impact of timeliness of acute stroke unit care; and (3) duration of hospitalization for stroke patients requiring rehabilitation increased over the course of the pandemic that could not be explained by any between-wave differences in stroke severity or functional dependency.

A drop in number of stroke patients admitted to hospitals has been reported frequently and was linked to an avoidance to seek hospital treatment during an increasing intensity of lockdown measures [11,12]. While conclusions are frequently drawn comparing stroke care during the pandemic with that one provided in a pre-pandemic setting [4,5,6,11,12], our data for comparing the first with the second wave of the pandemic supports the hypothesis that the number of stroke admissions largely depends on regional variability of SARS-CoV-2 incidence rather than strict containment measures alone. The site studied in this analysis was barely hit by the first wave of the pandemic, yet it belonged to the most severely affected regions in Germany during the second wave of the pandemic. Thus, increasing incidences of SARS-CoV-2 likely had a substantial effect on stroke patients’ or their relatives’ attitude toward health care utilization (e.g., because of fear of infection) and may partially explain the reduction in stroke admissions during the second wave of the pandemic. This hypothesis is further corroborated by our previous multi-center analysis of stroke and TIA admissions during the first wave of the pandemic [6]. While three study sites located in severely affected regions in Germany registered a remarkable decrease in admissions (up to 85%) during the first wave as compared with a pre-pandemic period, there was no such effect seen at the Dresden study site. Furthermore, incidence-rate ratio in our study was comparable to that one reported in an observational study of stroke admissions prior to and during the first pandemic months in the metropolitan area of Amsterdam that was severely impacted by COVID-19 [11]. Drop in stroke admissions in the current study, therefore, likely occurred in a time-staggered manner, according to increasing regional SARS-CoV-2 incidences.

Interestingly, we did not find any differences in stroke type or severity between both waves of the pandemic, suggesting a balanced drop of hospital admissions for the entire spectrum of stroke manifestations. Conversely, rates of both IVT and EVT were comparable between both waves and the pre-pandemic setting [8], thus indicating that patients potentially amenable to acute reperfusion therapies may still seek emergency medical care despite increasing SARS-CoV-2 incidence, also suggested by a recent nationwide analysis of 1463 hospitals in Germany [13]. Furthermore, a 13.2% decline of IVT and 12.7% of EVT during the first pandemic months was recently shown in an international retrospective study [4]. However, while 457 stroke centers from 70 countries worldwide contributed to this analysis, results are barely generalizable to individual stroke centers given the large variability of regional SARS-CoV-2 incidences, loco-regional allocation of resources, and infection control strategies. In addition, the decline in reperfusion rates observed in this global analysis could preferentially be a result of decreased hospital admissions and may not reflect quality of in-hospital hyperacute stroke care during the pandemic. In our study, IVT and EVT process times did not appear to be altered during the second wave, which is in line with our previous data from the first wave of the pandemic [8].

While key performance indicators of evidence-based hyperacute stroke care have been globally investigated in the context of the pandemic, data on the actual proportion of stroke patients with limited access to stroke unit care is sparse [4,6,11,14]. Intensification of in-hospital infection control measures, such as cohorting from the first to the second wave of the pandemic, likely had a substantial effect on patients receiving standardized acute stroke unit care at the same day of admission in our study. While only two cases of delayed stroke unit admissions in the first wave of the pandemic were possibly attributed to COVID-19 associated in-house relocations of resources (i.e., limited bed capacities), the majority of delayed stroke unit admissions were immediately related to COVID-19 (i.e., positively tested patients) in the second wave of the pandemic. This is of particular interest since key stroke outcomes, such as mortality, dependency, and institutional care, are substantially lower following comprehensive stroke unit care as compared to non-stroke unit care, and immediate hours and days following stroke are considered most critical in this matter [15,16,17]. Given the global burden of stroke, a protected stroke unit care area might constitute a model to be considered, especially for hospitals with high COVID-19 admissions and high-volume stroke centers.

Hospitalization was particularly prolonged in stroke patients discharged to a rehabilitation facility that could also have impacted hospital’s bed capacity, underscoring the importance of distributing acute, as well as rehabilitative, resources in a coordinated fashion across institutions. Preclinical studies have shown that efficacy of rehabilitation after stroke is time sensitive, with a decline in brain capacity for recovery associated with delayed rehabilitation [18]. Observations from studies performed in stroke patients also suggest a particular benefit from early rehabilitation starting within days, which is presumed to be evident also for broader time windows according to a monocentric case-control study performed in Italy [19,20].

Our study is limited by its relatively low sample size and monocentric design. In addition, our data does not allow any conclusions whether delayed access to stroke unit care may have had an adverse effect on functional and further stroke-specific outcomes. However, given the fact that timely stroke unit care constitutes an evidence-based therapy in acute stroke, its absence may serve as surrogate for potentially worse outcomes in these patients. Lastly, we did not implement data of the pre-pandemic era in this study precluding firm conclusions on a general impact of the COVID-19 pandemic on stroke care. However, this study was a follow-up of our previous findings, suggesting a relationship between drop in stroke admissions and local SARS-CoV-2 incidence during the first wave of the pandemic, but no relevant drop in reperfusion therapies [6].

## 5. Conclusions

While quality of care seems to be warranted in the hyperacute setting of stroke, subsequent stroke unit care might be particularly vulnerable to a detrimental impact of COVID-19. Protected code stroke protocols are an important measure to ensure standardized stroke care, but local strategies should also establish pathways for those requiring evidence-based stroke unit care despite an infection with SARS-CoV-2. Whether this substantiates the need for isolated stroke unit beds or entirely protected stroke units needs to be further elaborated in larger studies.

## Figures and Tables

**Figure 1 life-11-00710-f001:**
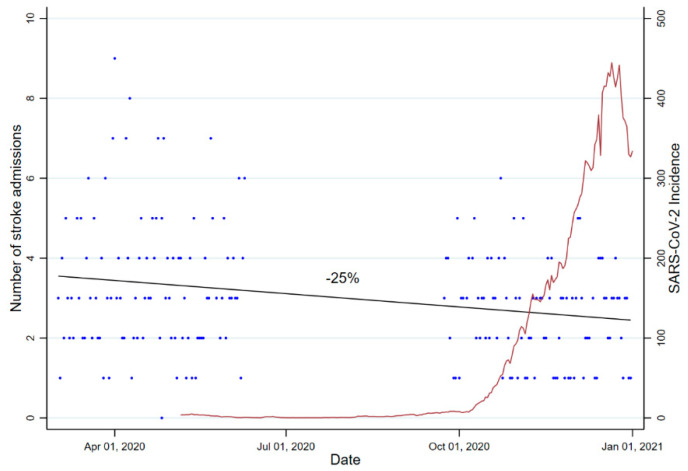
Number of stroke admissions per day, as well as the incidence of SARS-CoV-2, in Saxony. Per day are plotted. Black line constitutes an overlayed linear prediction plot for number of stroke admissions. Data on the incidence of SARS-CoV-2 in Saxony are reproduced from freely available data offered by RKI [10].

**Table 1 life-11-00710-t001:** Baseline characteristics of the study population.

Demographics	First Wave (n = 333)	Second Wave (n = 249)	*p*
Age, years, median (IQR)	77 (65,83)	78 (65,84)	0.65
Female, n (%)	153 (45.7)	109 (43.8)	0.65
Living alone, n (%)	102 (33.9)	81 (36.7)	0.52
Nursing home/care facility, n (%)	29 (8.1)	20 (8.1)	0.88
Care level/assistance needed, n (%)	109 (33.2)	71 (29.0)	0.32
**Comorbidities/patient history**			
Cardiovascular risk factors, n (%)			
Arterial hypertension	275 (82.3)	212 (85.5)	0.42
Diabetes mellitus	104 (31.1)	75 (30.1)	0.57
HbA1c in DM, median [IQR]	5.8 (5.5, 6.5)	5.8 (5.5, 6.6)	0.58
Hyperlipidemia	152 (45.5)	104 (41.9)	0.35
LDL-C in HLP, mean (±SD)	2.3 (1.0)	2.3 (1.1)	0.70
Obesity	111 (37.0)	91 (41.9)	0.27
Nicotine *	69 (33.2)	52 (28.6)	0.38
Coronary Heart disease	69 (20.8)	53 (21.4)	0.92
Atrial fibrillation	66 (19.9)	60 (24.1)	0.22
Cerebrovascular disease, n (%)	154 (46.8)	116 (46.9)	1.00
Ischemic stroke	153 (46.7)	111 (44.6)	
Hemorrhagic stroke	5 (1.5)	7 (2.8)	
Transitory ischemic attack **	4 (1.2)	1 (0.4)	
Further comorbidities, n (%)			
Deep venous thrombosis	9 (2.7)	7 (2.8)	1.00
Pulmonary embolism	7 (2.1)	5 (2.0)	1.00
Malignancy	52 (15.8)	36 (14.5)	0.73
Chronic lung disease	26 (7.9)	22 (8.8)	0.76
Dementia	33 (10.0)	19 (7.6)	0.38
Alcohol abuse ***	49 (26.6)	45 (31.3)	0.39
Psychiatric disorder	33 (10.0)	24 (9.7)	1.00

* Patients who quit smoking set as non-smoker in case they stopped at age <40 and for >20 years. ** In case of TIA and AIS, AIS was chosen. *** Yes, in case of >1 bottle of beer/day.

**Table 2 life-11-00710-t002:** Acute stroke and case management.

Acute Management	First Wave (n = 333)	Second Wave (n = 249)	*p*
SARS-CoV-2 positive at admission, n (%)	0 (0.0)	11 (4.4)	<0.0001
Delayed stroke unit admission, n (%)	5 (1.5)	17 (6.8)	0.001
Reperfusion therapy *, n (%)	122 (36.6)	93 (37.3)	0.86
Intravenous thrombolysis in-house	40 (12.1)	27 (11.0)	
Intravenous thrombolysis (outside)	34 (10.3)	21 (8.4)	
Endovascular therapy, n (%)	72 (21.7)	58 (23.4)	
Revascularization therapy, n (%)	25 (7.5)	13 (5.2)	0.31
Carotid endarterectomy	12 (3.6)	6 (2.4)	
Carotid Stenting	13 (3.9)	8 (3.2)	
**Process times ****			
Onset-to-admission (min), median (IQR)	90 (58.8, 202.8)	132 (64.8, 247.8)	0.98
Door-to-imaging (min) (IVT), median (IQR)	9 (6, 12)	12 (6, 21)	0.36
Door-to-needle (min), median (IQR)	38.4 (24.6, 53.4)	37.8 (25.8, 66)	0.47
Onset-to-groin (min), median (IQR)	234 (162, 300)	239 (187.2, 292.2)	0.70
Door-to-groin (min), median (IQR)	64.8 (51, 85.8)	70.2 (55.8, 97.8)	0.22
**Case management**			
Admission modality, n (%)			0.82
Via emergency medical service	188 (56.9)	142 (57.0)	
Intra-hospital	22 (6.7)	12 (4.9)	
Inter-hospital via SOS-NET	94 (28.5)	74 (29.7)	
Walk-in	26 (7.9)	21 (8.4)	
Discharge modality, n (%)			0.42
Home	150 (44.9)	110 (44.2)	
Rehabilitation center	118 (35.3)	87 (34.9)	
Nursing home	10 (3.0)	5 (2.1)	
Inter- and intrahospital	30 (9.0)	18 (7.2)	
Death	26 (7.8)	29 (11.6)	
Length of hospitalization (rehabilitation), days, median (IQR)	12 (8, 17)	20 (11, 27)	<0.0001
Length of hospitalization (excl. rehab), days, median (IQR)	6 (4, 10)	7 (5, 12)	0.01
Medication at discharge, n (%)			0.57
First ever prescription	129 (39.6)	87 (34.9)	
Dual antiplatelet therapy	72 (22.9)	45 (18.1)	
Direct oral anticoagulant	71 (22.6)	59 (23.7)	
Phenprocoumon	7 (2.2)	5 (2.0)	
LWMH/HWMH (therapeutic)	11 (3.5)	6 (2.4)	

* Constitutes IVT via telestroke network (SOS-NET) consultation or at index hospital and EVT at index hospital. ** Taking into account only patients admitted via emergency medical service. In cases last seen normal was known, mean duration (hours) was calculated, deviating cases were labeled as missing data. In case of admission via SOS-NET external door-to-needle times displayed. Door-to-imaging time not applicable for SOS-NET patients. Onset-to-groin time only displayed for cases with available exact onset time +/−1 h. In case of in hospital secondary deterioration time of deterioration set as “door time”.

**Table 3 life-11-00710-t003:** Vascular phenotypes.

Demographics	First Wave (n = 333)	Second Wave (n = 249)	*p*
Age, years, median (IQR)	77 (65,83)	78 (65,84)	0.65
Female, n (%)	153 (45.7)	109 (43.8)	0.65
Living alone, n (%)	102 (33.9)	81 (36.7)	0.52
Nursing home/care facility, n (%)	29 (8.1)	20 (8.1)	0.88
Care level/assistance needed, n (%)	109 (33.2)	71 (29.0)	0.32
**Comorbidities/patient history**			
Cardiovascular risk factors, n (%)			
Arterial hypertension	275 (82.3)	212 (85.5)	0.42
Diabetes mellitus	104 (31.1)	75 (30.1)	0.57
HbA1c in DM, median (IQR)	5.8 (5.5, 6.5)	5.8 (5.5, 6.6)	0.58
Hyperlipidemia	152 (45.5)	104 (41.9)	0.35
LDL-C in HLP, mean (±SD)	2.3 (1.0)	2.3 (1.1)	0.70
Obesity	111 (37.0)	91 (41.9)	0.27
Nicotine *	69 (33.2)	52 (28.6)	0.38
Coronary Heart disease	69 (20.8)	53 (21.4)	0.92
Atrial fibrillation	66 (19.9)	60 (24.1)	0.22
Cerebrovascular disease, n (%)	154 (46.8)	116 (46.9)	1.00
Ischemic stroke	153 (46.7)	111 (44.6)	
Hemorrhagic stroke	5 (1.5)	7 (2.8)	
Transitory ischemic attack **	4 (1.2)	1 (0.4)	
Further comorbidities, n (%)			
Deep venous thrombosis	9 (2.7)	7 (2.8)	1.00
Pulmonary embolism	7 (2.1)	5 (2.0)	1.00
Malignancy	52 (15.8)	36 (14.5)	0.73
Chronic lung disease	26 (7.9)	22 (8.8)	0.76
Dementia	33 (10.0)	19 (7.6)	0.38
Alcohol abuse ***	49 (26.6)	45 (31.3)	0.39
Psychiatric disorder	33 (10.0)	24 (9.7)	1.00

* For patients admitted via SOS-NET external initial NIHSS score displayed. ** Death resulted in NIHSS 42 points. *** Worsening defined as difference NIHSS (Discharge-Administration) >0.

## Data Availability

The data supporting the results of this study are available on reasonable request.

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
