# Peer review of "Delayed Comprehensive Stroke Unit Care Attributable to the Evolution of Infection Protection Measures across Two Consecutive Waves of the COVID-19 Pandemic"

_life, 2021, doi:10.3390/life11070710_

Round 1
Reviewer 1 Report
It is said that the COVID-19 pandemic affects access to healthcare for other diseases such as chronic health issues. The authors conducted very important research to prove this with evidence.
This study argues that stroke care has been delayed and patients did not have proper care on time due to the two waves of the COVID. However, this study did not show the situation before the COVID.
I suggest including the data of stroke care at this hospital before the COVID and show the different characteristics before and during the pandemic (e.g., reduced care, increased severity after COVID).
If it takes too much time to include all information, authors can include several measures related to stroke care and severity of symptoms/ or complications before the COVID waves briefly in the text.
During the pandemic, there were 878 patients admitted to the ER. An ER visit is an indicator of severe symptoms. For example, patients could not access the hospital properly, the ER visits could be increased compared to the pre-COVID period. If the authors show this kind of data, the conclusion can be stronger.
Author Response
It is said that the COVID-19 pandemic affects access to healthcare for other diseases such as chronic health issues. The authors conducted very important research to prove this with evidence.
Response: We would like to thank the reviewer for their kind comment on our study.
This study argues that stroke care has been delayed and patients did not have proper care on time due to the two waves of the COVID. However, this study did not show the situation before the COVID.
I suggest including the data of stroke care at this hospital before the COVID and show the different characteristics before and during the pandemic (e.g., reduced care, increased severity after COVID).
If it takes too much time to include all information, authors can include several measures related to stroke care and severity of symptoms/ or complications before the COVID waves briefly in the text.
Response: The reviewer made an excellent point. It is indeed of utmost interest how stroke care has changed between pre-pandemic and the pandemic era. However, there are several studies published with a large variability of results. While most of them suggest that stroke admissions largely decreased over the course of the first wave of the pandemic, only a few eventually show a decline in hyperacute stroke care measures (such as IVT/EVT frequencies and corresponding quality metrics). The authors of the submitted work also published several paper on this topic (e.g., Nogueira RG, Abdalkader M, Qureshi MM, et al. Global Impact of COVID-19 on Stroke Care and Intravenous Thrombolysis. Neurology, 2021; Barlinn K, Siepmann T, Pallesen LP, et al.. Universal laboratory testing for SARS-CoV-2 in hyperacute stroke during the COVID-19 pandemic. J Stroke Cerebrovasc Dis, 2020. 29(9): p. 105061; Hoyer C, Ebert A, Huttner HB, Puetz V, et al. Acute Stroke in Times of the COVID-19 Pandemic: A Multicenter Study. Stroke, 2020. 51(7): p. 2224-2227) that also have been cited and discussed in the submitted paper. Thus, in our study we primarily aimed to investigate whether the increase in SARS-CoV-2 incidence and intensified infection control measures had an impact on stroke admissions and in-hospital stroke care. For instance, while we did not find an impact of the pandemic on stroke and TIA admissions during the first wave of the pandemic, where SARS-CoV-2 incidence was very low at the authors’ study site (as shown in Hoyer C, Ebert A, Huttner HB, Puetz V, et al. Acute Stroke in Times of the COVID-19 Pandemic: A Multicenter Study. Stroke, 2020. 51(7): p. 2224-2227), admissions largely dropped during the second wave, when incidences strongly increased at the authors’ study site. This confirmed our previous hypothesis that such observations are not generalizable and rather are associated with local SARS-CoV-2 incidences. Another major finding was that intensified infection control measures applied in the second wave of the pandemic likely had a tremendous effect on stroke care (as reflected by the large number of patients deprived of acute stroke unit care within the first 24-hours after admission) as compared with the first wave of the pandemic. We therefore would like to omit additional data from the pre-pandemic era from the paper as we believe this would be beyond the scope of our manuscript.
However, as suggested by the reviewer we added a sentence to the text emphasizing this limitation and our previous work on this topic:
“Lastly, we did not implement data of the pre-pandemic era in this study precluding firm conclusions on a general impact of the COVID-19 pandemic on stroke care. However, this study was a follow-up of our previous findings suggesting a relationship between drop in stroke admissions and local SARS-CoV-2 incidence during the first wave of the pandemic, but no relevant drop in reperfusion therapies.[6]”
Data on the number of patients deprived of stroke unit care prior to the pandemic-era is unfortunately not available.
During the pandemic, there were 878 patients admitted to the ER. An ER visit is an indicator of severe symptoms. For example, patients could not access the hospital properly, the ER visits could be increased compared to the pre-COVID period. If the authors show this kind of data, the conclusion can be stronger.
Response: The reviewer made another excellent point. However, in Germany ER visits rather not correspond to severity of symptoms. Many patients are walk-ins suffering from minor symptoms (including headache, vertigo, sensory disturbances etc) of any cause. We are therefore not convinced that the suggested approach would eventually strengthen the conclusion of our paper. We hope the reviewer has an appreciation for our response.
Reviewer 2 Report
There is no data about cerebral venous thrombosis?
Also there is no data about atherosclerotic comorbodity as etiology os stroke and number isrwgeral.to thrombectomy, craniectomy and endarteromy
Author Response
There is no data about cerebral venous thrombosis?
Response: We would like to thank the reviewer for reviewing our manuscript and the valuable comments. We completely agree with the reviewer that any data on this topic would be of particular interest, especially in the context of COVID-19 vaccine-associated complications; however, there was no patient suffering from SVT during the study period.
Also there is no data about atherosclerotic comorbodity as etiology os stroke and number isrwgeral.to thrombectomy, craniectomy and endarteromy
Response: We thank the reviewer for this comment. We present a comprehensive overview of comorbidities, stroke etiologies and acute therapies including EVT and TEA in the Tables 1 and 2 of our submitted paper.